# The Function of Myostatin in Ameliorating Bone Metabolism Abnormalities in Individuals with Type 2 Diabetes Mellitus by Exercise

**DOI:** 10.3390/cimb47030158

**Published:** 2025-02-27

**Authors:** Chenghao Zhong, Xinyu Zeng, Xiaoyan Yi, Yuxin Yang, Jianbo Hu, Rongbin Yin, Xianghe Chen

**Affiliations:** 1College of Physical Education, Yangzhou University, Yangzhou 225009, China; 18950056613@163.com (C.Z.); zxy181101104@163.com (X.Z.); 15576961328@139.com (X.Y.); yyx781535082@163.com (Y.Y.); 18822098452@163.com (J.H.); 2School of Physical Education and Sport, Soochow University, Suzhou 215006, China; yinrongbin01@126.com

**Keywords:** myostatin, bone metabolism, exercise, type 2 diabetes mellitus

## Abstract

Purpose: The molecular mechanisms involved in bone metabolism abnormalities in individuals with type 2 diabetes mellitus (T2DM) are a prominent area of investigation within the life sciences field. Myostatin (MSTN), a member of the TGF-β superfamily, serves as a critical negative regulator of skeletal muscle growth and bone metabolism. Current research on the exercise-mediated regulation of MSTN expression predominantly focuses on its role in skeletal muscle. However, due to the intricate and multifaceted mechanical and biochemical interactions between muscle and bone, the precise mechanisms by which exercise modulates MSTN to enhance bone metabolic disorders in T2DM necessitate additional exploration. The objective of this review is to systematically synthesize and evaluate the role of MSTN in the development of bone metabolism disorders associated with T2DM and elucidate the underlying mechanisms influenced by exercise interventions, aiming to offer novel insights and theoretical recommendations for enhancing bone health through physical activity. Methods: Relevant articles in Chinese and English up to July 2024 were selected using specific search terms and databases (PubMed, CNKI, Web of Science); 147 studies were finally included after evaluation, and the reference lists were checked for other relevant research. Results: Myostatin’s heightened expression in the bone and skeletal muscle of individuals with T2DM can impede various pathways, such as PI3K/AKT/mTOR and Wnt/β-catenin, hindering osteoblast differentiation and bone mineralization. Additionally, it can stimulate osteoclast differentiation and bone resorption capacity by facilitating Smad2-dependent NFATc1 nuclear translocation and PI3K/AKT/AP-1-mediated pro-inflammatory factor expression pathways, thereby contributing to bone metabolism disorders. Physical exercise plays a crucial role in ameliorating bone metabolism abnormalities in individuals with T2DM. Exercise can activate pathways like Wnt/GSK-3β/β-catenin, thereby suppressing myostatin and downstream Smads, CCL20/CCR6, and Nox4 target gene expression, fostering bone formation, inhibiting bone resorption, and enhancing bone metabolism in T2DM. Conclusion: In the context of T2DM, MSTN has been shown to exacerbate bone metabolic disorders by inhibiting the differentiation of osteoblasts and the process of bone mineralization while simultaneously promoting the differentiation and activity of osteoclasts. Exercise interventions have demonstrated efficacy in downregulating MSTN expression, disrupting its downstream signaling pathways, and enhancing bone metabolism.

## 1. Introduction

Type 2 diabetes mellitus (T2DM) is a metabolic disorder characterized by persistently elevated blood glucose levels, primarily resulting from insulin resistance and inadequate pancreatic β-cell function [1]. An increasing body of evidence suggests that insulin resistance and the low-grade chronic inflammation induced by obesity are the primary causes of the onset and progression of T2DM [2,3]. According to estimates from the International Diabetes Federation (IDF), there were 537 million people living with diabetes worldwide in 2021, resulting in global healthcare expenditures of up to USD 966 billion. This figure is projected to exceed USD 1054 billion by 2045 [4]. In China, 52% of diabetes patients still do not receive effective treatment [5]. Patients with untreated T2DM frequently develop a range of complications, including renal, neurological, and cardiovascular diseases as well as bone-related conditions such as diabetic osteoporosis [1,6]. The non-enzymatic glycosylation of collagen under hyperglycemic conditions results in the production of advanced glycation end products (AGEs), which accumulate in bone tissue and subsequently compromise its microstructure and mechanical properties [7,8]. In T2DM patients, AGEs can interact with various cell-surface receptors including the receptor for advanced glycation end products, which triggers heightened intracellular inflammatory responses and oxidative stress [9,10]. These processes directly inhibit osteoblast-mediated bone formation while promoting osteoclast-mediated bone resorption [9,10]. Research indicates that individuals with prolonged T2DM, those undergoing insulin therapy, or those with poorly managed diabetes are at an increased risk of fractures [11,12]. Consequently, there is a pressing need to elucidate the mechanisms underlying bone metabolic disorders in T2DM and to develop effective treatment strategies.

Maintaining good glycemic control may help reduce the risk of bone metabolism disorders in patients with T2DM. Dietary intervention strategies have shown that vitamin D [13], high dietary fiber intake [14], the Mediterranean diet [14], and a balanced calcium-to-phosphorus ratio in the dietary structure [15] can lower the risk of bone metabolism disorders in T2DM. However, dietary strategies face compliance challenges and other side effects, such as hypercalcemia, renal impairment [16], and pesticide exposure [17]. The effects of various antidiabetic medications on bone metabolism are not uniform. Medications such as metformin and glucagon-like peptide-1 receptor agonists may provide a protective effect on bone metabolism or have no significant impact [18]. However, recent studies have identified age-specific effects [19] and renal hazards [20], while some medications exacerbate bone metabolism abnormalities, such as thiazolidinediones and canagliflozin [18]. Notably, T2DM-related bone metabolism disorders are often accompanied by a low bone-turnover state, making anti-resorptive medications such as bisphosphonates less than ideal choices [21]. Denosumab has been associated with bone loss following treatment discontinuation [22]. Abaloparatide has been linked to transient increases in heart rate and decreases in blood pressure [23]. In terms of metabolic intervention, metabolic surgery can significantly improve glucose metabolism, but rapid post-operative weight loss and nutritional malabsorption may lead to accelerated bone loss [24]. Similarly, substantial weight loss resulting from intensified lifestyle interventions may also pose an increased risk for fractures [25]. Currently, there is limited research directly comparing the efficacy of exercise with the aforementioned interventions for T2DM-related bone metabolism disorders. However, exercise has been shown to have a more favorable impact on bone metabolism than medication alone in obese populations [26] and in ovariectomized mice [27]. Furthermore, studies involving combined exercise and medication regimens indicate that exercise appears to provide a synergistic effect, protecting bone metabolism [28,29]. Compared with non-exercise therapies, exercise offers a safer and more sustainable intervention pathway.

Exercise is widely recognized as a crucial intervention for the prevention and management of T2DM [30]. It not only effectively regulates blood glucose levels [31] but also diminishes insulin resistance, improves body composition, and enhances endothelial function [32]. Furthermore, exercise is instrumental in managing chronic diseases [33], including diabetic osteoporosis [34]. Studies have demonstrated that bone quality, microstructure, and mechanical strength are significantly compromised in T2DM mouse models; however, these parameters improve with aerobic exercise and high-intensity interval training (HIIT) [35]. For instance, a study involving older adults with prediabetes revealed that 16 weeks of soccer training resulted in a 3.9% to 30.6% increase in bone mineral density in the femur and lumbar spine [36]. Additionally, previous research from our team indicated that downhill running mitigates abnormal osteoclast differentiation and bone resorption in T2DM mice, leading to an increase in bone mass [37].

Myostatin (MSTN), a member of the transforming growth factor β (TGF-β) superfamily, was first identified in 1997 [38]. This protein comprises 376 amino acid residues and exists initially as an inactive precursor that is activated through proteolytic cleavage [39]. The structural characteristics of MSTN include a signal peptide, a precursor region, and a mature C-terminal fragment, which is responsible for receptor binding and the execution of biological functions [39]. The role of MSTN in muscle metabolism has been extensively investigated, revealing that it is highly expressed in the skeletal muscle of both embryonic and adult mice [38], where it regulates muscle fiber number and type [40,41], inhibits muscle fiber growth, and, consequently, limits overall muscle mass increase [42,43,44,45]. Mice deficient in MSTN exhibit a marked increase in muscle mass [38]. As a principal negative regulator of muscle growth, strategies aimed at inhibiting MSTN have shown promise in the treatment of muscle atrophy, with several candidate drugs having successfully completed Phase II clinical trials and progressing to further development [46,47]. Moreover, MSTN is also implicated in bone metabolism; MSTN-deficient mice not only display increased muscle mass but also significant enhancements in bone density and trabecular architecture [48]. Administration of MSTN via intraperitoneal injection in mice results in decreased serum levels of Procollagen Type 1 N-Terminal Propeptide (P1NP) and elevated levels of cross-linked N-telopeptide (CTX), indicating suppressed bone formation and abnormally heightened bone resorption [49]. These findings suggest that MSTN plays a regulatory role in the differentiation and function of osteoblasts and osteoclasts, thereby influencing bone formation and resorption.

Importantly, various forms of exercise, including aerobic training (AT) [50,51,52] and resistance training (RT) [53], have been shown to reduce MSTN expression and ameliorate bone loss [54], indicating that MSTN may mediate the beneficial effects of exercise on bone metabolic disorders in T2DM. Presently, there exists a considerable body of research concerning the regulation of MSTN expression through exercise, predominantly centered on skeletal muscle. However, due to the intricate and multifaceted mechanical and biochemical interactions between muscle and bone, the precise mechanisms by which exercise modulates MSTN to enhance bone metabolic disorders in T2DM necessitate additional exploration. Consequently, the objective of this review is to systematically synthesize and evaluate the role of MSTN in the development of bone metabolism disorders associated with T2DM as well as to elucidate the underlying mechanisms influenced by exercise interventions. This analysis aims to offer novel insights and theoretical recommendations for enhancing bone health through physical activity.

## 2. Method

We conducted a narrative review and performed a non-systematic examination of the literature on the role of MSTN in exercise-induced improvements in bone metabolism disorders, particularly in the context of T2DM. To refine the focus of the review, we selected the most pertinent original research articles, clinical trials, meta-analyses, and reviews published in both Chinese and English up to July 2024 on this subject. The search terms included (individually and/or in combination) T2DM, diabetes, myostatin, exercise, mechanical force, diabetic bone disease, osteoporosis, bone metabolism, osteoblasts, osteoclasts, inflammation, and oxidative stress. For research purposes, we used PubMed, China National Knowledge Infrastructure (CNKI), and the Web of Science as electronic databases.

This study followed the PRISMA guidelines for systematic literature screening, with the specific process outlined as follows: A total of 379 potentially relevant articles were obtained through systematic searches. A three-tier screening mechanism was implemented based on predefined inclusion criteria. Initially, a title/abstract screening was conducted, with inclusion criteria consisting of (1) peer-reviewed original research or systematic reviews; (2) studies explicitly discussing the mechanisms of bone metabolism in T2DM; and (3) studies that included well-designed in vitro experiments, animal models, or clinical trials. During the initial screening, 183 articles were excluded, including duplicate publications (*n* = 13), non-research documents such as conference abstracts (*n* = 42), and articles with insufficient evidence levels, such as case reports (*n* = 128). Subsequently, the methodological quality of 196 studies was assessed, leading to the exclusion of 34 articles that had significant flaws in experimental design or lacked sufficient relevance to the mechanisms of T2DM bone metabolism. A full-text review was then conducted of the remaining 162 studies, with particular emphasis on excluding the following: (1) studies focusing on other diabetes-related complications (*n* = 49); (2) studies with subjects confounded by other metabolic diseases (*n* = 33); and (3) review articles and retracted publications (*n* = 33). Ultimately, 47 studies met all inclusion criteria and were included in the final analysis (Figure 1).

## 3. The Impact of MSTN on Bone Metabolism Disorder in T2DM

### 3.1. T2DM Exacerbates MSTN-Mediated Bone Metabolism Disorders

T2DM, as a complex metabolic disease, involves multiple processes such as insulin resistance, oxidative stress, and inflammation, which are interconnected and influence each other. In the context of T2DM, there are several mechanisms that further exacerbate MSTN-mediated bone metabolism disorders (Figure 2).

Insulin resistance represents a fundamental hallmark of T2DM characterized by diminished cellular responsiveness to insulin resulting in impaired glucose uptake and elevated blood glucose levels. This phenomenon disrupts normal cellular signaling pathways and undermines the organism’s ability to counteract oxidative damage, thereby creating an environment conducive to oxidative stress. Obesity is a significant contributor to insulin resistance; the excessive accumulation of adipose tissue induces adipocyte hypertrophy, which releases factors such as free fatty acids (FFAs) that disrupt insulin signaling pathways and reduce cellular sensitivity to insulin [55]. Under typical physiological conditions, the binding of insulin to its receptor activates the Phosphoinositide-3-kinase (PI3K)/Protein kinase B (AKT) signaling cascade. Specifically, insulin activates 3-Phosphoinositide-Dependent Protein Kinase 1 (PDK1), which subsequently phosphorylates AKT at the Thr308 residue. Further phosphorylation of AKT at the Ser473 site is required for its complete activation. The activated AKT phosphorylates Forkhead box O1 (FoxO1), leading to its exclusion from the nucleus and subsequent ubiquitination in the cytoplasm, ultimately resulting in its proteasomal degradation. This phosphorylation mechanism inhibits FoxO1 from translocating to the nucleus and binding to the MSTN promoter, thereby preventing the overexpression of MSTN. However, in the setting of insulin resistance, the PI3K/AKT/FoxO1 pathway becomes inactive, leading to elevated FoxO1 activity, increased MSTN expression, and the subsequent development of skeletal complications associated with T2DM. [56].

Oxidative stress is a pivotal factor in the pathogenesis of T2DM, arising from an imbalance between the generation of reactive oxygen species (ROS) and the antioxidant defense mechanisms. Hyperglycemia, a significant precipitating factor, facilitates the formation of AGEs and activates Protein Kinase C (PKC). The activation of the PKC pathway further stimulates the production of ROS, exacerbating oxidative stress [57]. Furthermore, the spontaneous oxidation of excess glucose can produce ROS, thereby exacerbating oxidative stress. Mitochondrial dysfunction is also a critical contributor to oxidative stress, as impaired electron transport chain function, dysregulation of mitochondrial uncoupling proteins (UCPs), and disruption of the coupling between electron transport and ATP synthesis lead to increased electron leakage, thereby intensifying oxidative stress [58]. Furthermore, research has indicated that the Nicotinamide adenine dinucleotide phosphate oxidase 4 (Nox4)-mediated signaling pathway is instrumental in the process by which MSTN inhibits bone formation [59]. Nox4 is also significantly involved in the differentiation of osteoblasts [60] and osteoclasts [61]. These findings imply that the downstream signaling pathways of MSTN may be influenced by the oxidative stress state induced by T2DM.

Inflammation is another critical component in the pathological progression of T2DM. In the context of obesity-related T2DM, there is an accumulation of macrophages within the adipose tissue and a state of chronic hypoxia, which leads to low-grade inflammation. This inflammatory response results in the release of pro-inflammatory cytokines such as interleukin-1 beta (IL-1β) and tumor necrosis factor-alpha (TNF-α), which interfere with insulin signaling through various mechanisms, thereby exacerbating insulin resistance [62,63]. Concurrently, the upregulation of MSTN expression mediated by these pro-inflammatory cytokines (TNF-α, IL-1, IL-17) further contributes to MSTN-related bone degradation [64]. Consequently, inflammation may serve as a critical factor in the disruption of bone metabolism mediated by MSTN in T2DM.

### 3.2. MSTN Regulates T2DM OB Differentiation and Bone Mineralization

Bone metabolism disorder in T2DM is characterized by reduced OB differentiation and diminished bone formation. In T2DM mice, there is a notable 9.4-fold increase in MSTN expression within the bones [65]. In a separate study, an 8-week treatment with MSTN antibodies in rats exhibited protective effects against the detrimental impacts of T2DM on femoral microstructure and the corresponding reduction in mechanical strength [66]. These findings indicate that MSTN plays a regulatory role in T2DM osteoblasts. Furthermore, numerous investigations have employed MSTN to treat primary osteoblasts from mice [49], primary bone marrow-derived mesenchymal stem cells (BMSCs) from mice [67], and the human fetal osteoblast cell line 1.19 [68], resulting in a noted downregulation of gene expression specific to osteoblasts. MSTN exhibits a dose-dependent inhibitory effect on the activity of alkaline phosphatase (ALP) and the secretion of osteocalcin in osteoblasts. Additionally, it suppresses the expression of osteogenic factors in BMSCs, including Osterix, Runt-related transcription factor 2 (Runx2), osteoblast-specific factor 2 (OSF-2), bone morphogenetic protein-2 (BMP-2), and insulin-like growth factor 1 (IGF-1). This suppression leads to a reduction in osteoblast differentiation and bone mineralization, while simultaneously promoting adipogenic differentiation [49,67,68]. Upon investigating its mechanism, it was found that the MSTN receptor Activin receptor IIB (ActRIIB) is present on the membranes of osteoblasts, fibroblasts, chondrocytes, and BMSCs [69]. MSTN binds to ActRIIB and forms a heteromeric complex with Activin receptor-like kinase 4 (ALK4) and ALK5, leading to the phosphorylation of the highly conserved and membrane-associated I-type receptors (ALK4, ALK5). Activated I-type receptors phosphorylate the cell signaling molecules Smad2 and Smad3, which subsequently combine with Smad4 and translocate to the nucleus, thereby inhibiting osteoblast differentiation [70,71]. The fundamental mechanism underlying MSTN-mediated osteoblast differentiation in T2DM is attributed to the modulation of key osteogenic genes via the Smad signaling pathways.

MSTN has been shown to influence the process of bone mineralization. Specifically, treatment with MSTN resulted in a reduction in citrate secretion by BMSCs during the osteogenic differentiation in the C3H10T1/2 cell line [59]. IGF-1 facilitates the release of citrate by BMSCs via the PI3K/AKT/Mammalian target of rapamycin (mTOR) pathway to activate rat sarcoma (Ras) for regulation [72]. MSTN, on the other hand, can impede this pathway and hinder BMSCs from secreting citrate. Nevertheless, by suppressing the function of Nox4 in BMSCs, the suppressive impact of MSTN on citrate secretion is counteracted, thereby reducing its inhibitory effect on OB differentiation [59]. Nox4-induced ROS contribute to oxidative stress in both the early and chronic stages of T2DM [73]. Additionally, research has indicated that Nox4 regulates osteoblasts via the Mitogen-activated Protein Kinase (MAPK) signaling pathway or promotes osteoblastogenesis through the TGF-β pathway [74,75]. This suggests that Nox4 may modulate T2DM OB through various pathways. UMR106 rat osteoblast-like cells exhibit a direct upregulation of fibroblast growth factor 23 (FGF23) expression following treatment with MSTN. Upon binding to ActRIIB, MSTN activates the intracellular p38 MAPK/nuclear factor-kappa B (NF-κB) signaling pathway. This activation subsequently enhances Store-operated Ca^2+^ entry (SOCE), leading to an increase in FGF23 expression. The resulting FGF23 can then interact with fibroblast growth factor receptor 3 (FGFR3) in either an autocrine or paracrine fashion [76]. This activation results in the activation of the extracellular regulated protein kinases 1/2 (ERK1/2), leading to the transcriptional suppression of tissue-nonspecific alkaline phosphatase (TNAP) [77]. The reduced TNAP activity on the cell membrane inhibits the conversion of Pyrophosphoric acid (PPi) to inorganic phosphate (Pi), ultimately impairing bone mineralization [78].

MicroRNAs play a crucial role in maintaining bone homeostasis by modulating the gene expression of osteoblasts [79]. Specifically, microRNA-218 (miR-218) is involved in the regulation of the Wnt signaling pathway, as it inhibits the activity of sclerostin (SOST) and Dickkopf-2 (DKK2) during osteoblast differentiation [80]. In Ocy454 cells exposed to MSTN a notable downregulation of miR-218 expression was detected. This alteration not only impacts the intracellular levels of miR-218 but also results in a decrease in its presence within exosomes. Furthermore, exosomes altered by MSTN have been shown to impede osteoblast differentiation when co-cultured with MC3T3 cells. Specifically, the heightened SOST and DKK1 levels impede the activity of the transcription factor Tcf7 and facilitate the phosphorylation of Glycogen synthase kinase 3 beta (GSK-3β). This phosphorylation triggers the degradation of β-catenin, consequently inhibiting Wnt pathway activity [81]. The Wnt signaling pathway plays a crucial role in transducing mechanical forces into biochemical signals within osteocytes. Mechanical stimulation triggers the activation of AKT, which subsequently facilitates the phosphorylation of GSK-3β via primary cilia, integrins, prostaglandin E2 (PGE2), and/or the FoxO signaling pathway. This process enhances the expression of Wnt proteins and activates the Wnt/β-catenin pathway, either within the same cells or in adjacent cells [82]. A reduction in the activity of this pathway is ultimately associated with the development of osteoporosis in the bone tissue of individuals with T2DM. While MSTN-altered exosomes may not directly impact OC, they can induce bone cells to release the receptor activator for nuclear factor-κB ligand (RANKL), thereby indirectly modulating OC function and bone resorption [81]. Furthermore, MSTN competitively obstructs the TGF-β1 pathway mediated by ALK5, impeding the upregulation of SRY-box transcription factor 9 (Sox9) in chondrogenesis. This inhibition interferes with the chondrogenic differentiation of BMSCs and the proliferation of chondrocytes, consequently exerting a negative influence on bone formation [83]. The involvement of miRNAs in the regulation of bone homeostasis has been the subject of considerable research. MSTN and its associated signaling pathways play a critical role in maintaining the equilibrium between bone formation and resorption. Consequently, the interplay between miRNAs and MSTN may offer valuable insights for developing therapeutic strategies for T2DM.

### 3.3. MSTN Regulates T2DM OC Differentiation

The extent of bone resorption in individuals with T2DM may be underestimated, a situation attributed to reduced collagen cross-linking facilitated by lysyl oxidase in T2DM. This reduction may result in potentially flawed methodologies for detecting cross-linked peptides that are released during the bone resorption process [84,85]. Given the overexpression of MSTN in T2DM and its stimulatory effect on bone resorption, further research is essential to understand the influence of MSTN on osteoclasts within the context of bone metabolic disturbances associated with T2DM. In the process of differentiating mouse primary bone marrow-derived macrophages (BMMs) into osteoclasts, MSTN exhibits elevated expression levels in both osteoclast precursors and mature osteoclasts; however, it is absent in BMMs. This expression is stimulated by RANKL during the initial phases of differentiation [64], indicating that MSTN may serve as a specific regulatory factor for OC. In vitro studies have shown that the administration of MSTN results in an increased number of tartrate-resistant acid phosphatase (TRAP)-positive multinucleated osteoclasts along with enhanced TRAP activity in bone marrow mononuclear cells and BMMs. This phenomenon is observed to be dose-dependent and is associated with the upregulation of nuclear factor of activated T-cell cytoplasmic 1 (NFATc1), thereby facilitating the process of osteoclastogenesis [64]. Mechanistically, MSTN binds to the heterodimeric receptor composed of ActRIIB and ALK4/5 on the surface of the OC precursors and mature OCs, activating ALK4/5, releasing FK506-binding protein 12 (FKBP12) and methylated Smad7, exposing the substrate binding site of Smad2, and facilitating the recruitment of R-Smad2 [86]. This activation leads to the phosphorylation of Smad2, its binding to NFATc1 [64], and its translocation into the nucleus of OC precursors, thereby upregulating the expression of integrin αv, integrin β3, Dendritic cell-specific transmembrane protein (DC-STAMP), and calcitonin receptor, which promotes OC differentiation [64]. In a recent investigation, researchers administered MSTN to RAW264.7 cells and discovered that MSTN is capable of modulating the RANKL-activated NF-κB and MAPK signaling pathways via Smad2. This modulation promotes the Smad2-dependent nuclear translocation of NFATc1, which subsequently leads to the upregulation of genes implicated in osteoclast differentiation, thereby enhancing the differentiation process of osteoclasts [64]. During this process, the expression of the Coiled-coil domain-containing protein 50 (Ccdc50) gene is notably reduced, and its overexpression can counteract the stimulative effect of the NF-κB and MAPKs pathways on MSTN’s function in OC generation [87]. Furthermore, in comparison with the co-culture of wild-type (Wt) osteoblasts and Wt BMMs, the co-culture of MSTN−/− osteoblasts and Wt BMMs resulted in a 90% reduction in the number of osteoclasts generated [64]. This finding suggests that MSTN functions as both an autocrine and paracrine regulatory factor in osteoclastogenesis, being secreted by both osteoblasts and BMMs. This effect may be attributed to MSTN’s capacity to enhance the expression of RANKL mRNA in osteocytes, thereby increasing the transport of soluble RANKL protein to osteoclasts and indirectly promoting osteoclast activity, which facilitates their differentiation [81]. The mechanisms underlying the interactions of MSTN between these two cell types warrant further investigation. In conclusion, the activation of the NF-κB and MAPK pathways is modulated by the Smad2 protein, thereby linking the MSTN signaling pathway with the classical osteoclast activation pathway. Although prior research has examined MSTN derived from osteoclasts, osteoblasts, and myocytes, our comprehension of the specific mechanisms and the differential roles of MSTN from various tissue sources in bone metabolism remains insufficiently developed.

Metabolic dysregulations such as lipotoxicity, oxidative stress, and insulin resistance can instigate localized inflammation in the pathogenesis of T2DM, thereby influencing the onset and progression of the disease [88]. Furthermore, research indicates that plasma levels of MSTN are significantly elevated in individuals diagnosed with T2DM [89] and show a positive correlation with the levels of pro-inflammatory cytokines [90]. In vitro studies have demonstrated that the stimulation of human primary synovial fibroblasts with recombinant inflammatory cytokines such as TNF-α, IL-1, and IL-17 results in the upregulation of MSTN expression [64]. Furthermore, in cultured muscle cells, TNF-α enhances MSTN expression via an NF-κB-dependent mechanism, which subsequently promotes IL-6 expression through the p38 MAPK and Mitogen-activated protein kinase (MEK1) signaling pathways [91]. Notably, MSTN has also been shown to upregulate TNF-α. In MH7A cells treated with MSTN, TNF-α expression increases in a concentration-dependent manner. This suggests that MSTN may directly influence TNF-α production by binding to the ALK receptor and activating the PI3K/AKT/AP-1 signaling pathway [92]. Additionally, MSTN treatment of MH7A cells may directly activate the c-Jun N-terminal kinase (JNK)/ERK/AP-1 pathway and downregulate miR-21-5p, thereby enhancing IL-1β expression [93]. In experimental models involving the injection of TNF-α into mouse cranial bones, an upregulation of RANKL expression was observed. Subsequent studies utilizing TNF-α to treat cultured bone cells revealed that TNF-α can induce RANKL expression by modulating MAPK phosphorylation and activating NF-κB [94]. The effects of TNF-α are analogous to those of RANKL and occur independently of RANKL; it sequentially activates NF-κB p50/p52 and c-fos, which, as downstream signaling molecules, bind to the NFATc1 promoter to enhance NFATc1 expression levels, thereby facilitating osteoclast differentiation [95]. Moreover, TNF-α pre-activated osteoclast precursors produce IL-1β in an autocrine manner through interactions with bone matrix proteins, promoting the maturation of osteoclasts from these precursors to resorb bone [94]. Furthermore, MSTN treatment of fibroblast-like synovial cells induces the migration of T-helper 17 (Th17) cells to inflamed tissues. This process is mediated by MSTN’s upregulation of chemokine ligand 20 (CCL20) expression via the Smads pathway, with CCL20 facilitating this process by binding to chemokine receptor 6 (CCR6) [96]. In conclusion, MSTN exerts inhibitory effects on osteoblast differentiation and function through various mechanisms while simultaneously promoting osteoclast differentiation and bone resorption (Figure 3).

## 4. The Role of MSTN in Improving Bone Metabolism Disorders in T2DM by Exercise

Currently, there is limited human research regarding the improvement of bone metabolism disorders in T2DM through exercise. However, studies have shown that activities such as soccer [36] and intermittent exercise [97] can effectively enhance skeletal health, as evidenced by improvements in bone mineral density and serum bone turnover markers (including osteocalcin, CTX, and P1NP). Notably, exercise interventions can also help maintain bone mass stability during weight loss in T2DM patients [98]. In animal studies, exercise has been shown to improve bone microstructure through multidimensional mechanisms, optimizing biomechanical properties [99,100,101,102]. It activates the osteogenic differentiation regulatory network (including Osterix, BMP-2, and ALP) via pathways such as autophagy [100], Wnt3a/β-catenin [103], and pyroptosis [104], thereby remodeling the RANKL/osteoprotegerin (OPG) balance axis [99,100,101,102]. This multifaceted targeting approach aims to repair bone metabolism disorders associated with T2DM. Nevertheless, there is still a lack of direct evidence indicating whether muscle factors (such as myostatin) mediate the effects of exercise on bone metabolism, which limits a comprehensive understanding of the underlying mechanisms of the exercise effects.

### 4.1. Role of MSTN in Ameliorating T2DM Bone Metabolism Disorders by Exercise

In individuals diagnosed with T2DM and obesity, there is a significant increase in MSTN levels in the skeletal muscle, bone tissue, and plasma [105]. The application of MSTN inhibitors or the genetic knockout of the MSTN gene has been shown to confer protection to murine models against the detrimental effects on bone microstructure and strength that are typically associated with T2DM [106]. Research has shown that both a 12-week combined training (CT) program [107] and a 12-week resistance training (RT) program [108] effectively reduce MSTN levels in the serum and plasma of individuals with T2DM. Additionally, an 8-week regimen of AT and HIIT has been shown to significantly ameliorate the adverse effects of T2DM on femoral bone mass, trabecular microstructure, cortical bone geometry, and overall bone mechanical strength in murine models [35]. Furthermore, following a 6-week weight-bearing running training protocol, streptozotocin (STZ)-induced diabetic rats exhibited a reduction in the bone resorption marker TRAP and an elevation in the bone formation marker ALP in their serum alongside a decrease in the expression levels of ActRIIB and Smad2/3 [109]. This indicates that exercise can reduce MSTN expression and the components of its typical signaling pathway, thereby enhancing bone metabolism in individuals with T2DM. The underlying molecular mechanism involves the suppression of MSTN, resulting in the deactivation of the ActRIIB/Smad2/3 signaling pathway and the activation of the Wnt/GSK-3β/β-catenin signaling pathway. GSK-3β serves as a negative regulatory factor in T2DM bone metabolism, and its excessive activity may aggravate osteoporosis and increase fracture risk [110]. Weight-bearing running has been shown to downregulate MSTN, thereby promoting the expression of Wnt and β-catenin in the femur of diabetic rats, reducing GSK-3β expression and enhancing T2DM bone metabolism [109]. Furthermore, a study involving resistance training with ten male participants revealed a significant increase in the concentration of Decorin (DCN) in skeletal muscle alongside a notable decrease in the concentrations of both serum and skeletal muscle MSTN. This phenomenon may be attributed to DCN inhibiting the activity of MSTN through its binding to the immobilized extracellular matrix (ECM) MSTN while simultaneously enhancing the expression of several MSTN inhibitors, including follistatin, which serves to obstruct MSTN’s entry into the circulatory system [111]. Furthermore, the regulatory influence of moderate exercise on matrix metalloproteinases 2 (MMP-2) and tissue inhibitors of metalloproteinases 2 (TIMP-2) may have contributed to the synthesis of DCN [112]. Moreover, research has demonstrated a downregulation of MSTN expression in response to resistance training in both animal and human subjects that is associated with the activation of the Notch signaling pathway [113]. The initiation of Notch signaling is a complex process that involves the cleavage of the transmembrane receptor, resulting in the formation of the Notch intracellular domain (NICD). Once translocated to the nucleus, the NICD interacts with CSL (a collective term for C promoter-binding factor 1, Hairless, and Lag1), thereby facilitating the upregulation of the Notch target gene expression [114]. Additionally, studies have indicated that low-intensity exercise in murine models leads to an increase in Androgen receptor (AR) levels, which in turn suppresses MSTN transcription by diminishing the expression of CCAAT/Enhancer binding protein δ (C/EBPδ), thereby further inhibiting MSTN-induced inflammatory responses [115]. These findings suggest that exercise may enhance bone metabolism in individuals with T2DM through the modulation of MSTN expression levels. Furthermore, a study demonstrated that treatment with a mitochondrial open reading frame of the 12S ribosomal RNA type-c (MOTS-c) in C2C12 mouse myoblasts upregulated the activity of casein kinase 2 (CK2). CK2 inhibits the activity of phosphatase and tensin homolog (PTEN) by phosphorylating specific amino acid clusters at the C-terminal (including Ser380, Thr382, Thr283, and Ser385), creating conditions for the activation of AKT. Meanwhile, MOTS-c may regulate mTORC2 activity by upregulating SIN1, which increases the phosphorylation of AKT and subsequently phosphorylates FoxO1. Phosphorylated FoxO1 translocates from the nucleus to the cytoplasm, where it is ubiquitinated and degraded, preventing its binding to the MSTN gene promoter and thereby downregulating MSTN expression [116]. Other studies have indicated that aerobic exercise in skeletal muscle activates AMP-activated protein kinase (AMPK) and its downstream target peroxisome proliferator-activated receptor-gamma coactivator-1 alpha (PGC-1α), which increases the expression of MOTS-c and regulates bone metabolic balance. This suggests that exercise may modulate the expression of MSTN by upregulating MOTS-c, thereby improving bone metabolism in patients with T2DM [56].

### 4.2. Role of the MSTN Signal Pathway in Ameliorating T2DM Bone Metabolism Disorders by Exercise

The exacerbation of oxidative stress in T2DM can be attributed to its pathological characteristics, which lead to the overproduction of ROS and a disruption of the body’s antioxidant defense mechanisms. In the context of T2DM, MSTN has the capacity to stimulate the release of ROS through the upregulation of NADPH oxidase and the activation of the ERK pathway [117]. Nox4, recognized as a pivotal enzyme in the production of ROS, also plays a significant role in mediating the inhibition of the IGF-1 signaling pathway by MSTN, which subsequently results in compromised bone mineralization [118]. These findings underscore the significant involvement of MSTN and Nox4 in the regulation of oxidative stress and bone metabolism abnormalities in T2DM. In conclusion, Nox4 may play a role in mediating the adverse effects of MSTN on bone metabolism in individuals with T2DM. However, the interactions between Nox4 and MSTN, along with the precise mechanisms through which exercise enhances bone metabolism in T2DM, remain to be elucidated. Studies have revealed high expressions of Nox4 in pre-osteoblasts and mature osteoblasts, where it acts as a crucial enzyme catalyzing ROS production during BMSCs differentiation [60]. Moreover, Nox4 is implicated in the RANKL-mediated abnormal enhancement of osteoclast differentiation in T2DM through mitochondrial ROS [119], which serve as secondary messengers in various signaling pathways such as MAPK, NF-κB, and Ca^2+^, ultimately upregulating NFATc1 expression to facilitate osteoclast differentiation and function [120,121]. While initial research highlighted the role of Nox4 in regulating osteoblast and osteoclast activities [59,60], subsequent studies have demonstrated that elevated Nox4 expression promotes the differentiation of BMSCs into adipocytes rather than osteoblasts [122,123]. Obesity affects the bone marrow microenvironment through the Nox4–ROS pathway, which accelerates the adipogenic differentiation of BMSCs at the expense of osteogenic differentiation. This shift may contribute to increased bone fragility in individuals with obesity-related T2DM [60]. Research conducted in both animal models and human subjects has demonstrated that physical exercise can mitigate the formation of ROS and alleviate Nox4-mediated oxidative stress by enhancing mitochondrial function [124,125,126,127,128]. Furthermore, the silencing of Nox4 in murine mesenchymal stem cell lines has been shown to counteract the inhibitory effects of MSTN on osteoblast differentiation and functionality, thereby promoting bone formation [59]. Additionally, downregulation of Nox4 expression in osteoclasts derived from the RAW264.7 mouse macrophage cell line as well as in p91 knockout mice has been found to decrease superoxide production and its subsequent stimulatory effect on bone resorption [129,130].

MSTN may mediate the improvement of inflammatory responses through exercise. Following a wheel-running exercise intervention in ovariectomized mice, CD8+ T cells (CD8Ts) were activated, leading to the secretion of Interferon-gamma (IFN-γ). This secretion subsequently inhibited the NF-κB signaling pathway (specifically, the inhibitor of κB (IκB) and p65) as well as the MAPK pathway (including ERK and p38) during the osteoclastogenesis process induced by BMMs [131]. This mechanism results in a reduced expression of NFATc1, which may subsequently disrupt the MSTN-mediated Smad2-dependent nuclear translocation of NFATc1. This disruption is associated with the upregulation of genes that play a role in osteoclast differentiation [64]. The reduced expression of NFATc1 consequently results in the downregulation of osteoclast marker genes, including cathepsin K (CTSK), MMP-9, CTR, and tumor necrosis factor receptor-associated factor 6 (TRAF6). This sequence of events ultimately inhibits osteoclastogenesis and ameliorates abnormal bone resorption [131]. A recent investigation demonstrated that MSTN enhances the expression of IL-6 in C2C12 myotubes in a dose-dependent manner. Notably, when C2C12 myotubes were co-cultured with TNF-α, there was a significant upregulation of MSTN expression [91]. In the context of STZ-induced diabetic rats, AT was found to ameliorate MSTN overexpression by decreasing the serum levels of inflammatory markers including IL-1β, IL-6, and TNF-α [132]. Furthermore, AT was shown to mitigate the upregulation of MSTN expression mediated by TNF-α through the activation of the NF-κB and p38 MAPK pathways in C2C12 myoblasts [131]. Conversely, other studies have indicated that exercise can lead to an increase in IL-6 levels. One particular study employed electrical pulse stimulation on the C2C12 mouse skeletal muscle cell line to mimic mechanical stimulation, revealing that skeletal muscle contraction induces intracellular Ca2+ release, which subsequently activates p38 MAPK and calcineurin (CaN). This activation triggers the JNK/AP-1 pathway in contracting muscles, resulting in the upregulation of IL-6 expression [133]. The intracellular signaling pathway associated with IL-6 was initially characterized in monocytes and macrophages exposed to lipopolysaccharide (LPS). In these immune cells, LPS engages myeloid differentiation primary response 88 (MyD88) through Toll-like receptor 4 (TLR-4), initiating a signaling cascade involving interleukin-1 receptor-associated kinase-1 (IRAK-1) and TRAF6, which activates the IκB kinase (IKK)/NF-κB pathway [134]. The activation of IKK results in the phosphorylation of IκB, rendering it susceptible to ubiquitination and subsequent proteasomal degradation. This process activates NF-κB, facilitating its translocation to the nucleus and promoting the transcription of immune-related genes, including IL-6, TNF-α, and IL-1β, thereby orchestrating the typical inflammatory response [134]. Notably, the upregulation of IL-6 induced by exercise does not correspond with an increase in IL-1β or TNF-α levels. While IL-6 elicits a pro-inflammatory response in monocytes and macrophages via the NF-κB pathway, the contraction-induced IL-6 signaling in muscle cells operates independently of NF-κB. This signaling promotes macrophages to produce the interleukin-1 receptor antagonist (IL-1ra) and IL-10, which inhibit IL-1β signaling and TNF-α expression, respectively [135,136,137], thereby facilitating an anti-inflammatory response associated with exercise. This mechanism may further inhibit MSTN through the modulation of IL-1β and TNF-α-mediated osteoclast differentiation as well as TNF-α-dependent MSTN expression via the NF-κB pathway [91]. In summary, the interplay among MSTN, exercise, pro-inflammatory cytokines, and bone metabolism in T2DM constitutes a complex feedback loop centered on MSTN. This network regulates both MSTN expression and the levels of pro-inflammatory cytokines, ultimately contributing to the amelioration of bone metabolic disorders in patients with T2DM (Figure 4).

## 5. Discussion

Existing research has established that, in the context of T2DM, there is an overexpression of MSTN, which leads to the dysregulation of key genes involved in bone metabolism. Exercise may alleviate bone metabolic disorders by inhibiting the overexpression of MSTN, obstructing its binding to targets, and disrupting its downstream signaling pathways (Figure 5).

The examination of the pathophysiology associated with musculoskeletal complications in T2DM highlights the significance of MSTN as a critical area of research. Nonetheless, it is imperative to also consider the contributions of other myokines and their interactions with MSTN in the context of bone metabolism disorders that arise as complications of diabetes affecting the musculoskeletal system. Exercise training is anticipated to mitigate the onset of diabetic myopathy and diabetic bone disease, thereby conferring protective benefits to both muscle and bone tissues. It is crucial to delineate the distinct types of exercise and their respective mechanisms of action. In addition to MSTN, Irisin, another myokine secreted by skeletal muscles, plays a vital role in the pathophysiology of musculoskeletal complications in T2DM. Irisin facilitates the development of brown adipose tissue and enhances energy expenditure, which in turn improves insulin sensitivity and addresses bone metabolism disorders [138]. Research suggests that exercise-induced skeletal Irisin can mitigate diabetes-related bone loss through the miR-150-FNDC5/pyroptosis axis. Specifically, exercise-mediated skeletal Irisin not only counteracts the hyperglycemia and hyperinsulinemia associated with T2DM but also enhances glucose tolerance in T2DM murine models by modulating the miR-150-FNDC5/Irisin signaling pathway [104]. Furthermore, combined aerobic and resistance training regimens have been shown to elevate serum Irisin levels in T2DM patients, with a noted negative correlation between Irisin and MSTN levels [107]. Studies indicate that the absence of MSTN activates PGC-1α, promoting Irisin secretion and facilitating the browning of white adipocytes [139]. Furthermore, Irisin exhibits a negative correlation with MSTN, fasting plasma glucose (FPG), and triglyceride (TG) levels, and there exists an independent relationship between Irisin concentrations and MSTN levels [140]. In individuals with T2DM, circulating Irisin levels are significantly lower compared with those in healthy individuals [141]. Notably, physical exercise has been shown to enhance Irisin levels through the upregulation of FNDC5 expression [142]. The increased expression of Irisin subsequently promotes the expression of glucose transporters, mitochondrial biogenesis, and the differentiation of brown adipocytes, which collectively facilitate thermogenesis and lipid metabolism. This process operates in conjunction with MSTN, targeting similar pathways to ameliorate bone metabolic disorders associated with T2DM [140]. Although the current literature indicates a reduction in Irisin levels among T2DM patients, there is substantial variability in the findings across studies, which may be attributed to differences in detection methodologies [143].

Moreover, FGF21 has been identified as a factor that can enhance insulin sensitivity and bone metabolism [144]. Research has linked FGF21 expression to mitochondrial dysfunction and stress in skeletal muscle. Deficiencies in autophagy lead to mitochondrial dysfunction, which subsequently elevates FGF21 levels, thereby aiding in the prevention of diet-induced obesity and insulin resistance while promoting muscle mass and function [145,146]. MSTN modulates the expression of Kruppel-like factor 4 (KLF4) and FGF21 through the Smad2/3 and p38 signaling pathways in adipocytes. A reduction in KLF4 and FGF21 levels can result in mitochondrial dysfunction and inflammation due to MSTN deficiency [147]. DCN, a small proteoglycan secreted by skeletal muscle during contraction, has been shown to promote muscle growth. In a study utilizing low-dose streptozotocin and a high-fat diet to induce diabetic cardiomyopathy, DCN therapy was evaluated, revealing that DCN overexpression exacerbated cardiac functional impairment but effectively mitigated fibrosis and inflammation through the insulin-like growth factor 1 receptor (IGF1R)/PKCα/Heat shock protein 70 (HSP70) and TGFβ1 pathways [148]. Additionally, DCN inhibits the anti-myogenic effects of MSTN by binding to and inactivating it. Overexpression of DCN enhances the expression of the myogenic factor Mighty, while also increasing the expression of Myogenic differentiation 1 (Myod1) and follistatin and decreasing the expression of atrophy-related F-box protein (atrogin1) and Muscle RING finger 1 (MuRF1) [144]. Consequently, DCN may function as a myogenic factor and presents potential as a therapeutic target for addressing muscle atrophy or inflammation associated with T2DM.

Research has shown that AT is associated with reduced fasting insulin levels and Homeostasis Model Assessment (HOMA) scores in overweight and obese children and adolescents, potentially preventing metabolic syndrome and T2DM [149,150]. RT, classified as anaerobic exercise, increases muscle fiber number and volume, thereby enhancing the body’s insulin response and improving glucose transport [151]. Excessive MSTN expression inhibits muscle cell proliferation and differentiation [152], whereas RT can downregulate MSTN expression, facilitating muscle growth and repair [53]. Furthermore, the combined effects of AT and RT yield greater benefits than either form of exercise alone [153]. Additional research suggests that a CT approach is more effective than exclusive aerobic training in significantly reducing fasting blood glucose (FBG), HbA1c (glycated hemoglobin), and HOMA-IR levels, thereby improving insulin resistance markers in overweight or obese children and adolescents [153,154,155]. Different exercise modalities exert varying effects on MSTN levels. Some studies have demonstrated that aerobic exercise can lower MSTN concentrations; for instance, a specific duration of AT has been shown to reduce plasma and muscle MSTN levels in healthy or overweight/obese individuals and T1DM rats [156,157]. RT may also influence MSTN, potentially through pathways associated with muscle hypertrophy and strength development, although the precise effects are protocol dependent [158]. In a study involving male T2DM patients, serum MSTN concentrations significantly decreased following 12 weeks of CT [107].

The influence of MSTN on bone metabolism disorders in T2DM is significantly impacted by the type of exercise performed [158,159]. Research indicates that AT indirectly inhibits MSTN expression by modulating endocrine functions (such as lowering blood glucose and insulin levels) and enhancing energy expenditure, which reduces fat accumulation and alleviates systemic inflammation and insulin resistance, thereby promoting glucose uptake by skeletal muscles and insulin secretion by the pancreas, ultimately improving bone metabolism disorders in T2DM patients [153,156]. A comparative analysis of various exercise types on bone changes in T2DM revealed that HIIT and AT can reverse certain adverse alterations in cortical and trabecular bone, while swimming may exacerbate these changes, and RT can lead to femoral bone deformation. Notably, HIIT significantly decreased the yield stress and failure load capacity of the femur, while aerobic exercise was correlated with a relative increase in the expression of Osterix, Runx2, and ALP [35]. Furthermore, our previous research indicated that swimming training significantly increased Runx2 mRNA expression but did not activate the Wnt3a/β-catenin pathway compared with downhill running, nor did it significantly enhance trabecular morphology and bone histomorphometric parameters of the distal femur [103]. Swimming training also significantly elevated the RANKL/OPG ratio, suggesting that while it may promote early osteogenic differentiation factors, it could also lead to bone loss due to insufficient gravitational stimulation [102]. The effects of RT on bone metabolism in T2DM may vary across different anatomical sites. A 6-week RT program effectively increased bone density in the proximal tibial metaphysis of T2DM rats and improved bone quality [99], while RT reduced cortical bone thickness in T2DM mice, resulting in bone deformation [35]. Recent studies have also indicated that varying intensities of AT (such as low versus high intensity) exhibit distinct anti-inflammatory mechanisms that enhance bone metabolism in T2DM. The effects of different intensities of AT on bone morphology and biomechanical properties are also divergent; low-intensity AT improves trabecular quality by mitigating inflammation and oxidative stress in bone tissue, while higher-intensity AT programs enhance cortical bone density and fracture resistance by reducing hyperglycemia and inflammation in bone tissue [160]. Our previous research demonstrated that an 8-week HIIT regimen could promote bone formation in T2DM mice via the Meg3/P62/Runx2 pathway [100]. Consistent with these findings, a 6-week HIIT intervention increased both cortical bone density and humeral bone density in T2DM rats [161]. In conclusion, aerobic exercise and HIIT may represent superior strategies for addressing bone metabolism disorders in T2DM.

Notably, recent investigations have introduced an innovative in vitro model to explore the interplay between muscle and skeletal systems in T2DM. This study developed modular skeletal muscle constructs utilizing cross-linked gelatin hydrogels and fibrin hydrogels to assess the effects of endurance training and HIIT on MSTN expression under hyperglycemic conditions [162]. This system offers a personalized microenvironment, potentially aiding in the identification of relevant therapeutic targets for diabetic bone disease.

Currently, there is a lack of sufficient research on the enhancement of bone metabolism in T2DM through MSTN in sports, with several unresolved issues that warrant further investigation: (1) MSTN can be produced by muscles, transported to target sites via the bloodstream, or released by bone tissues to impact bones directly. Exploring the effects and roles of MSTN from various sources on bone metabolism is crucial for comprehending its mode of operation. (2) It is imperative to enhance the understanding of the molecular regulatory mechanism network of MSTN on bone metabolism in T2DM across different exercise modalities, intensities, frequencies, and durations. This will facilitate the development of evidence-based exercise prescription strategies and establish a robust theoretical foundation for exercise intervention studies. (3) MSTN, acting as a secretory regulatory factor originating from osteoblasts and bone marrow macrophages, could offer potential therapeutic targets for addressing bone degradation by elucidating the interplay between MSTN, osteoblasts, and osteoclasts. (4) The synergistic or antagonistic effects between MSTN and other factors affecting bone metabolism, such as Nox4 and FGF21, also need to be further understood.

## 6. Conclusions

In the context of T2DM, MSTN has been shown to exacerbate bone metabolic disorders by inhibiting the differentiation of osteoblasts and the process of bone mineralization while simultaneously promoting the differentiation and activity of osteoclasts. Exercise interventions have demonstrated efficacy in downregulating MSTN expression, disrupting its downstream signaling pathways, and enhancing bone metabolism. Various forms of exercise, including aerobic activities and resistance training, exhibit differential impacts on MSTN levels and bone metabolism in individuals with T2DM. Furthermore, other myokines, such as Irisin and FGF21, are also recognized for their significant roles in the pathology of T2DM. However, the interactions between muscular and skeletal systems warrant further investigation to establish a clearer understanding.

## Figures and Tables

**Figure 1 cimb-47-00158-f001:**
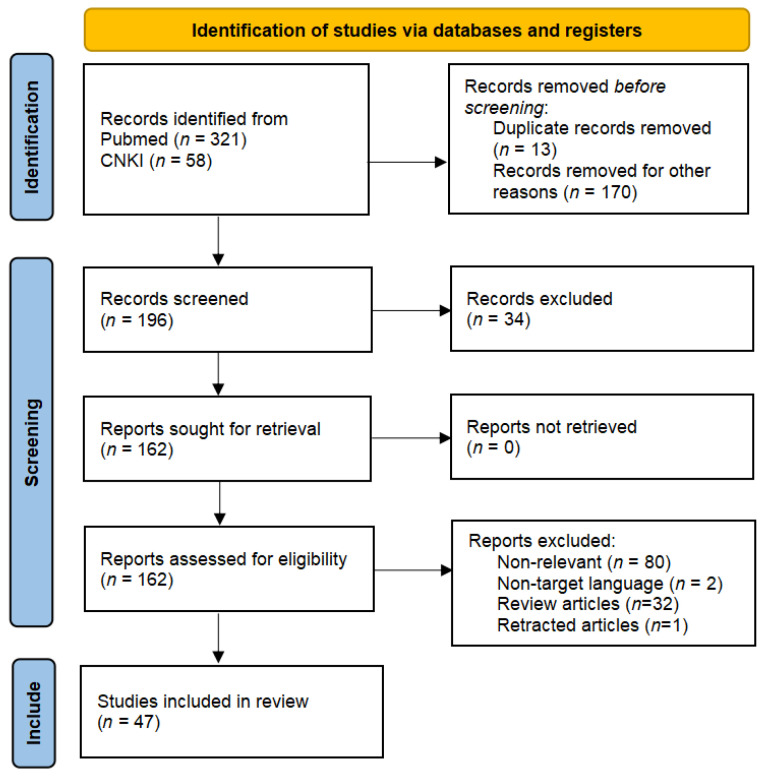
The PRISMA 2020 flow diagram was used for the identification of the studies included in this review. No automation tools were used for the screening process.

**Figure 2 cimb-47-00158-f002:**
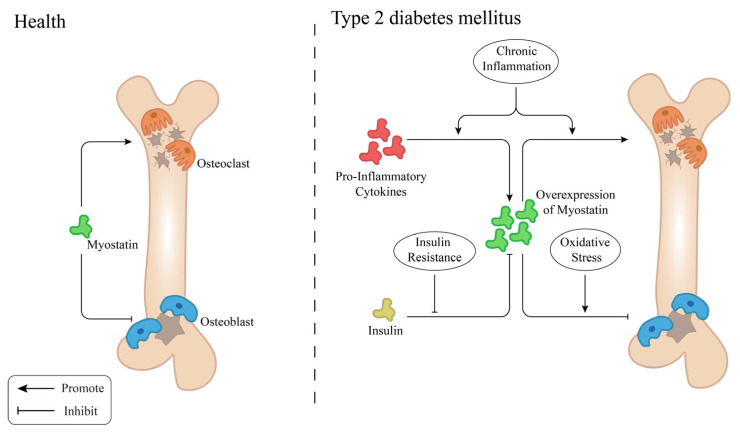
The function of MSTN in bone metabolism related to health and T2DM. Generated with Adobe Illustrator 2023 v27.0.

**Figure 3 cimb-47-00158-f003:**
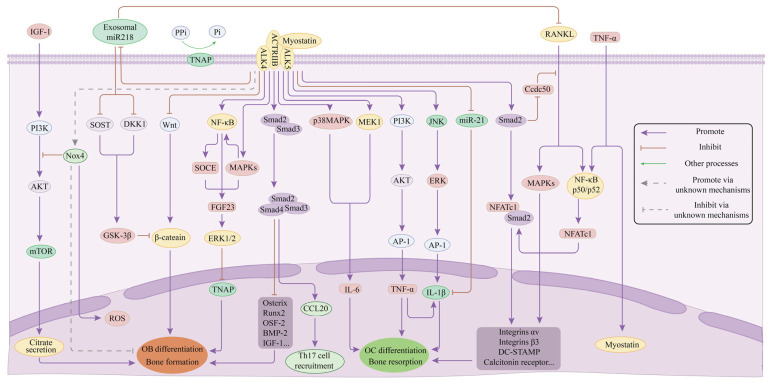
Mechanism of MSTN in bone metabolism disorder of T2DM. IGF-1—insulin-like growth factor 1; PI3k—Phosphoinositide-3 kinase; AKT—protein kinase B; mTOR—Mammalian target of rapamycin; NOX4—Nicotinamide adenine dinucleotide phosphate oxidase 4; SOST—sclerostin; DKK1—Dickkopf-1; GSK3β—Glycogen synthase kinase 3 beta; ROS—reactive oxygen species; NF-κB—nuclear factor-kappa B; SOCE—Store-operated Ca^2+^ entry; MAPK—Mitogen-activated protein kinase; FGF23—fibroblast growth factor 23; ERK1/2—extracellular regulated protein kinases 1/2; TNAP—tissue-nonspecific alkaline phosphatase; PPi—Pyrophosphoric acid; Pi—inorganic phosphate; ALK4—Activin receptor-like kinases 4; ALK5—Activin receptor-like kinases 5; ACTRIIB—Activin receptor IIB; Runx2—Runt-related transcription factor 2; OSF-2—osteoblast-specific factor 2; BMP-2—bone morphogenetic protein-2; MEK1—Mitogen-activated protein kinase 1; IL-6—interleukin-6; AP-1—activator protein 1; TNF-α—tumor necrosis factor-alpha; JNK—cJun N-terminal kinase; NFATc1—nuclear factor of activated T cells 1; Ccdc50—Coiled-coil domain-containing protein 50; RANKL—receptor activator for nuclear factor-κB ligand. MSTN inhibits OB differentiation and bone mineralization through multiple signal pathways, promoting OC differentiation and bone resorption, leading to disturbed bone metabolism in T2DM. Generated with Adobe Illustrator 2023 v27.0.

**Figure 4 cimb-47-00158-f004:**
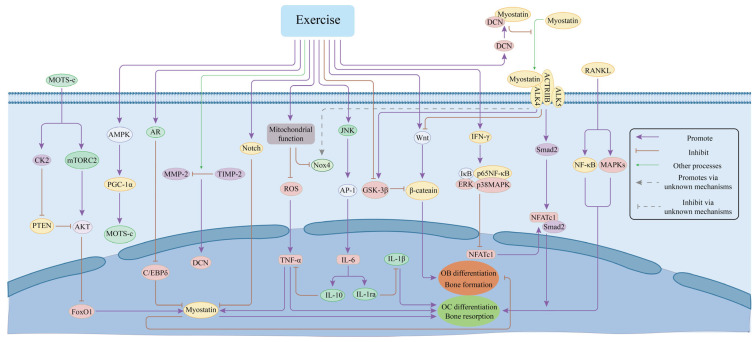
Mechanisms of MSTN in the improvement of T2DM bone metabolism disorders by exercise. MOTS-c—mitochondrial open reading frame of the 12S ribosomal RNA type-c; CK2—casein kinase 2; mTORC2—Mammalian target of rapamycin complex 1; PTEN—phosphatase and tensin homolog; AMPK—AMP-activated protein kinase; PGC-1α—peroxisome proliferator-activated receptor-gamma coactivator-1 alpha; AR—Androgen receptor; C/EBPδ—CCAAT/Enhancer binding protein δ; MMP-2—matrix metalloproteinases 2; TIMP-2—tissue inhibitor of metalloproteinases 2; DCN—Decorin; ROS—reactive oxygen species; TNF-α—tumor necrosis factor-alpha; JNK—cJun N-terminal kinase; AP-1—activator protein 1; IL-6—interleukin-6; IL-1β—interleukin-1 beta; IL-1ra—interleukin-1 receptor antagonist; GSK3β—Glycogen synthase kinase 3 beta; IFN-γ—Interferon-gamma; IκB—inhibitor of κB; NF-κB—nuclear factor-kappa B; ERK—extracellular regulated protein kinases; MAPK—Mitogen-activated protein kinase; NFATc1—nuclear factor of activated T cells; ALK4—Activin receptor-like kinases 4; ALK5—Activin receptor-like kinases 5; ACTRIIB—Activin receptor IIB; RANKL—receptor activator for nuclear factor-κB ligand. Exercise inhibits MSTN expression directly through multiple signal pathways or inhibits MSTN downstream pathways by improving mitochondrial function, inflammatory response, etc., which in turn improves T2DM bone metabolism disorders. Generated with Adobe Illustrator 2023 v27.0.

**Figure 5 cimb-47-00158-f005:**
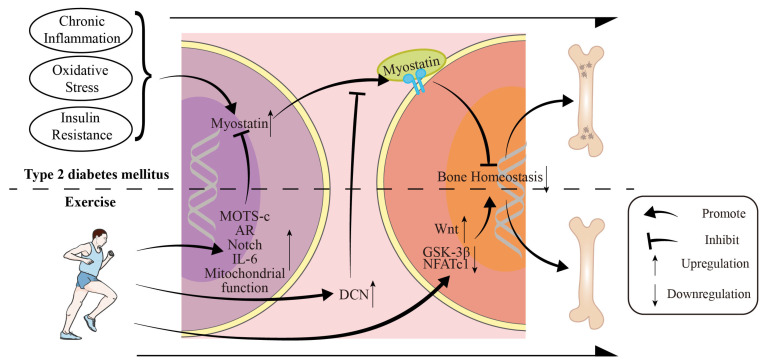
Schematic diagram of how exercise improves myostatin-mediated bone metabolism disorders in T2DM. The part above the dashed line indicates how MSTN impairs bone metabolism in the T2DM state, while the part below the dashed line indicates how exercise reverses this process. MOTS-c—mitochondrial open reading frame of the 12S ribosomal RNA type-c; AR—Androgen receptor; DCN—Decorin; IL-6—interleukin-6; GSK3β—Glycogen synthase kinase 3 beta; NFATc1—nuclear factor of activated T cells. Generated with Adobe Illustrator 2023 v27.0. Runner icon by Servier (https://smart.servier.com/ (accessed on 15 October 2023)) is licensed under CC-BY 3.0 Unported (https://creativecommons.org/licenses/by/3.0/).

## Data Availability

No new data were created.

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
