# Peer review of "The Function of Myostatin in Ameliorating Bone Metabolism Abnormalities in Individuals with Type 2 Diabetes Mellitus by Exercise"

_cimb, 2025, doi:10.3390/cimb47030158_

Round 1
Reviewer 1 Report
Comments and Suggestions for Authors
This review summarized and systematically evaluated the role of Myostatin in the development of metabolic bone disorders associated with type 2 diabetes. The authors described how physical exercise can influence these disorders improving bone health.
In my opinion the topic is interesting and relevant in the field of bone disorders and diabetes. The manuscript is well written and well-structured, and the sections shortly present available literature data.
I only have some minor observations:
Line 13: please add the acronym MSTN after “Myostatin”
Method: please add a flowchart to better describe the articles selection criteria used.
Figures 2 and 3: is it possible to use a bigger font?
Reviewer 2 Report
Comments and Suggestions for Authors
Thank you for the opportunity to review the paper submitted by Zhong et al to Current Issues in Molecular Biology. Please see below my comments on the above manuscript:
1-Line 14: please define MSTN
2-Line 62-73: before starting on the benefits of exercise, it’s important to add a paragraph about other possible solutions such ad dietary, pharmaceutical, etc. for improving the complications (focusing more on diabetic osteoporosis) related with diabetes and then mention why this paper is focused on exercise among all possible interventions.
3-Line 118-123: what was the selection criteria and/or narrowing down 379 articles to 147 studies used? Please describe the step-by-step process for going down from 379 to 196 and then 196 to 162 and finally 162 to 147. What factors were used for exclusion?
4-Line 131: If any type of program was used to create the Fig. 1 and other Figs that needs to be mentioned in the legends.
5-Line 354-513: prior to this section, that would be important to review the effect of exercise on bone health during diabetes. Once we get some background information on that, then the role of MSTN on bone metabolism disorders in diabetes as regulated by exercise can be discussed (section 4)
6-Line 638-648: please include some insights on future directions and the areas that need more research for both basic scientists and clinicians.
Author Response
|
Comments 1: 1-Line 14: please define MSTN |
|
Response 1: Thank you for the reviewer's guidance. In the revised manuscript, we have added the abbreviation "MSTN" when Myostatin is first mentioned in the abstract and provided a brief introduction to ensure that readers can clearly understand its meaning. Please refer to line 13-14 for details. |
|
Comments 2: 2-Line 62-73: before starting on the benefits of exercise, it’s important to add a paragraph about other possible solutions such ad dietary, pharmaceutical, etc. for improving the complications (focusing more on diabetic osteoporosis) related with diabetes and then mention why this paper is focused on exercise among all possible interventions. |
|
Response 2: Thank you for the valuable comments from the reviewer. Based on your suggestions, we have added a section in the introduction discussing other potential solutions for bone metabolism disorders in T2DM before addressing the benefits of exercise. This includes dietary interventions, T2DM medications, anti-osteoporosis drugs, and lifestyle modifications. We have detailed the potential benefits and drawbacks of these options and discussed why this paper specifically focuses on exercise as an intervention, outlining its advantages and disadvantages as well. Please refer to line 64-91 for details. The modified sections are marked in red. |
|
Comments 3: 3-Line 118-123: what was the selection criteria and/or narrowing down 379 articles to 147 studies used? Please describe the step-by-step process for going down from 379 to 196 and then 196 to 162 and finally 162 to 147. What factors were used for exclusion? |
|
Response 3:Thank you for the detailed information you provided. Based on your description, we have clarified the literature selection process in the manuscript to better reflect the rigor and transparency of the study. Below is the literature selection process for this study: A total of 379 potentially relevant articles were obtained through systematic searches. A three-level screening mechanism based on predefined inclusion criteria was implemented: First, a preliminary screening of titles/abstracts was conducted, which included the following inclusion criteria: (1) peer-reviewed original research or systematic reviews; (2) studies that explicitly explore the mechanisms of bone metabolism in type 2 diabetes mellitus (T2DM); (3) studies with robust methodological designs including in vitro experiments, animal models, or clinical trial designs. A total of 183 articles were excluded during the preliminary screening, including duplicates (n=13), non-research articles such as conference abstracts (n=42), and case reports with insufficient evidence levels (n=128). Subsequently, 196 studies underwent a methodological quality assessment, resulting in the exclusion of 34 articles due to significant design flaws or insufficient relevance to the mechanisms of T2DM bone metabolism. The remaining 162 studies were subjected to a full-text review, with a focus on excluding the following types of literature: (1) studies focusing on other diabetes-related complications (n=49); (2) studies with subjects confounded by other metabolic diseases (n=33); (3) review articles and retracted publications (n=33). Ultimately, 47 studies met all inclusion criteria and were included in the final analysis. We will add a description of the above literature selection process to the manuscript and include a PRISMA flow diagram to help readers clearly understand the sources and selection criteria of the study. Thank you again for your detailed explanations and suggestions. Please refer to line 152-172 for details. The modified sections are marked in red. |
|
Comments 4: 4-Line 131: If any type of program was used to create the Fig. 1 and other Figs that needs to be mentioned in the legends. |
|
Response 4: Thank you for your valuable comments. Figures 2, 3, 4, and 5 in the revised manuscript were created using Adobe Illustrator 2023 v27.0. Additionally, in Figure 5 of the revised manuscript, the runner icon by Servier (https://smart.servier.com/) is licensed under CC-BY 3.0 Unported (https://creativecommons.org/licenses/by/3.0/). Based on your feedback, we reviewed the citation practices for software in published papers and have added the relevant information in the figure captions. Please refer to line 181,405,588,601 for details. The modified sections are marked in red. |
|
Comments 5: 5-Line 354-513: prior to this section, that would be important to review the effect of exercise on bone health during diabetes. Once we get some background information on that, then the role of MSTN on bone metabolism disorders in diabetes as regulated by exercise can be discussed (section 4) |
|
Response 5: Thank you very much for your valuable suggestions. We fully agree that it is important to review the impact of exercise on bone health in diabetes patients before discussing "the role of MSTN in exercise regulation of bone metabolism disorders in diabetes." We have added a paragraph at the beginning of Section 4 that summarizes the effects of exercise on bone health in patients with T2DM, based on findings from human and animal studies. This provides background information for the subsequent exploration of the role of MSTN in exercise-regulated bone metabolism disorders in T2DM. Please refer to line 409-424 for details. The modified sections are marked in red. |
|
Comments 6: 6-Line 638-648: please include some insights on future directions and the areas that need more research for both basic scientists and clinicians. |
|
Response 6: Thank you for the constructive feedback from the reviewers. Based on your suggestions, we have added a forward-looking discussion about future research directions at the end of the discussion section of the article. Please refer to lines 721-735for details. The modified sections are marked in red. |
|
4. Response to Comments on the Quality of English Language |
|
5. Additional clarifications |
|
Dear reviewers and editors, Thank you very much for the constructive feedback on our research. We have made several additional modifications in the manuscript: 1. In the Methods section, we discovered an error; the references we ultimately included in the analysis are not 147, but 47. We appreciate the reviewer’s suggestion that brought this to our attention, and we have made the necessary correction in the manuscript. Please refer to line xx for details. The modified section is marked in red. 2. In the Discussion section, the runner icon in Figure 5 was provided by Servier. The relevant copyright information is as follows: "Runner icon by Servier https://smart.servier.com/ is licensed under CC-BY 3.0 Unported https://creativecommons.org/licenses/by/3.0/." We have included this information in the legend of Figure 5. Please refer to line xx for details. The modified section is marked in red. 3. Rreference number 40 in the original manuscript has been retracted, and we have removed it from the revised manuscript. Thank you for your consideration. |
Round 2
Reviewer 2 Report
Comments and Suggestions for Authors
The authors have done a great job addressing all of my concerns and have applied the necessary changes in the revised manuscript.